# Synergistic Effect between China's Digital Transformation and Economic Development: A Study Based on Sustainable Development

**Min Zhao [1], Rong Liu [2,\*] and Debao Dai [2]**

[1]   SILC Business School, Shanghai University, Shanghai 201800, China; zhaomin1226@163.com
[2]   Management School, Shanghai University, Shanghai 200444, China; ddb@shu.edu.cn
*    Correspondence: liurong0428@163.com; Tel.: +86-1853-672-1428

**Abstract:** Developing rapidly over the long term makes it easy for a developing country to fall into the middle-income trap, which can only be solved by a new technological revolution. The deep integration of digital technology and industry has gradually become a new impetus to promote the sustainable development of China's economy. Based on the panel data of 30 provinces and cities from 2010 to 2019, this paper analyzes the coupling coordination relationship between digital transformation level and economic development in China by the entropy method, coupling coordination degree model and spatial autocorrelation model. The results show that the following: (1) from 2010 to 2019, the comprehensive index of China's digital transformation and economic growth level showed an upward trend, and the development level showed a gradual decline in eastern–middle–western regions; (2) the level of coupling and coordination between China's digital transformation and economic growth has been increasing each year. Except Guangdong Province, all provinces have shown digital lag coordinated development, and it is necessary to strengthen their economic sustainability; (3) the coupling and coordination degree of digital transformation and economic growth in China shows a remarkable spatial correlation and agglomeration. High–high agglomeration areas are mainly concentrated in the Beijing–Tianjin–Hebei and Yangtze River Delta regions, low–low agglomeration areas are concentrated in northeast and western regions, and low–high agglomeration areas and high–low agglomeration areas are concentrated in southeast provinces. It is suggested that China should strengthen its regional balance in the future, let digital technology continue to lead the development of eastern developed regions, and transform and promote the traditional economy in other regions, overtaking in corners and enhancing the sustainable development of the overall layout of China's economy.

**Keywords:** digital transformation; economic development; coupling and coordination

## 1. Introduction

China's economy has developed rapidly, but the traditional extensive development has led to a series of environmental problems such as resource consumption, environmental pollution and ecological destruction, which is not conducive to the sustainable development of the economy. If the traditional rapid growth mode is left unchecked, the Chinese economy will easily fall into the middle-income trap. At present, all countries aim to achieve sustainable development and pursue sustainable economic development with the premise of ensuring that the ecological environment is not destroyed. With the emergence and application of artificial intelligence, big data, the Internet of Things, and other digital technologies, China's digital transformation has become a new direction of economic development. Many studies have pointed out that the emergence of digital technology can improve the utilization rate of resources and reduce the consumption of resources and environmental pollution with the premise of maintaining stable economic development [1–3]. Habanik thinks that, to realize the sustainable economic and social

development of countries and regions, we should focus on the application of digital technology in all fields of society [4]. The research of Jovanovi and Jiao pointed out that the level of digitalization is related to economic development. In countries with a high degree of digitalization, their economic development and social sustainable development are better [5,6].

Transformation refers to a series of economic activities with "internet plus" as the carrier, digitalization as a new factor of production, and the effective use of information and communication technology as a means of improving efficiency and optimizing the economic structure [7]. At present, digital transformation is involved in all aspects of daily production and life, such as digital transformation for consumers and digital transformation for industry, including industrial industry, retail industry, service industry, and government services. The report of the 19th National Congress of the Communist Party of China pointed out that a "Digital China" should be built to promote the deep integration of the internet, big data, artificial intelligence, and the real economy. The rapid development of the digital economy has empowered enterprises to carry out digital transformation. As the micro foundation of economic development, enterprises need to make substantial changes in their economic mode through transformation and upgrading [8]. As an important direction of future development, digitalization is the main driving force for economic growth. On the one hand, digital transformation has changed the business model and operation process of enterprises. Vaska and Rachinger, after analyzing the influence of digital technology on business model innovation, believe that digital transformation has affected the value creation, delivery, and acquisition of almost every industry, and new business models such as the sharing economy and circular economy can all be a driving force for sustainable development [9,10]. Warner thinks that digital transformation exists in the operation of industry companies as a continuous process [11]. Han and Vahid, respectively, explored the impact of digital transformation on value creation from the perspective of digital technology development and technology market expansion [12,13].

On the other hand, digital transformation promotes industrial upgrading and industrial integration. Liu thinks that the deep integration of the digital economy and real economy has promoted the revitalization and transformation of real industry [14]. The continuous application of digital technology in the manufacturing process has a great impact on the activities of production, organization management, and business management in the manufacturing industry and is further developing towards intelligence [15]. The integration of digital technology and traditional industry mainly includes two aspects: on the one hand, the underlying technology, intelligent production equipment, production system and industrial internet produced by information network technology; on the other hand, information technology is applied to various fields and levels, including product informatization, value chain informatization, and business model and enterprise operation internetization [16,17]. Chen et al. tested the spatial benefits of digital transformation on the development of industrial integration through the Cobb Douglas function and concluded that all input elements of digital transformation can promote the development of industrial integration [18]. By promoting the transformation and upgrading of the industrial structure and industrial technology innovation, digital technology creates higher economic value and greater enterprise competitiveness, which is leading China's industrial economy to form a new pattern of sustainable development.

The research shows that digital transformation promotes the growth of China's value chain. Ma et al. believe that the digital transformation of the production mode will have a great impact on the global industrial division of labor, the value chain, trade and investment, and then affect the whole economic system [19]. Ding uses the input–output data of WIOD to prove that the input of the digital economy can promote the growth of the domestic value-added rate of manufacturing exports and effectively promote the optimization and upgrading of China's export trade structure [20]. Guo found through research that the ICT (Information and Communication Technology) industry can promote industrial upgrading and increase the added value of traditional industries [21]. Zhang et al. also proved

through empirical analysis that digital investment can significantly improve the division of labor position of enterprises in the global value chain and is the value growth point of manufacturing development [22]. Qiu also believes that digital economy can help small and medium-sized enterprises to reduce costs, create new value and boost the enterprise value chain upgrade [23].

Therefore, the digital empowerment of the manufacturing industry is an urgent requirement for economic development [24]. Yi et al., based on the survey data of Chinese enterprises in 2012, tested the impact of digital transformation on enterprises' exports and proved that digital transformation is beneficial to expanding enterprises' exports [25]. Liu used the method of text mining to obtain data about the digital transformation of Chinese listed manufacturing companies and explored the direct and indirect effects of digital transformation on improving manufacturing productivity [26]. Zhang built an econometric model based on the panel data of 30 cities in China from 2015 to 2019 to verify the impact mechanism of the digital economy on high-quality economic development [27]. Chi et al., based on the survey data, conducted an empirical test with the research idea of "Digital Transformation—R&D Dual Capability—New Product Development Performance", and the results proved that the digital transformation of SMEs is beneficial to the improvement of new product development performance [28]. Ribeiro studied the impact of digital transformation on enterprise performance in the knowledge-intensive service industry, and the results showed that the application of digital technology can significantly promote an enterprise's financial performance [29]. Katz measured the impact of digitalization on GDP development through an endogenous growth model, and the research results showed that there was a significant positive impact [30].

In summary, the existing research mainly focuses on the impact and contribution of digital transformation on economic growth. As for how economic development can boost digital transformation, the existing research mainly considers the financial support for the digital economy from four aspects—consumption, production, market, and industrial development—and points out that economic development can lead to digital infrastructure construction, digital market construction, digital talent training, and digital platform development for digital development [31].

From the above documents, we can find that digital transformation and economic development interact with each other. On the one hand, digital transformation effectively boosts economic development. The remarkable feature of digital transformation lies in the application of digital technology to practical scenes such as enterprise production management, social activities, and government governance. Through digital technology, the rational allocation of production factors such as manpower and capital can be realized, and the total factor productivity can be improved. In addition, digital technology makes all kinds of information tend to be transparent. Under this condition, enterprises can reduce the information transaction costs and realize accurate matching between supply and demand. In social activities, the emergence of digital consumption scenes such as online transactions and mobile payment has further improved the economic vitality. In addition, government departments have improved the relevant government services through digital transformation, which not only reduces the management cost but also improves management efficiency.

On the other hand, economic development promotes digital transformation. Digital development needs information infrastructure construction and capital investment, as well as high-tech and high-level talents. The continuous improvement of the level of economic development means that the economy is more capable of providing these material bases. They are mutually coordinated and complementary; thus, the contribution of this paper is the study of the sustainable and coordinated development of digital transformation and economic growth.

In summary, the existing literature focuses on the influence of digital transformation on economic development, and most of these works use regression analysis and other methods to study and explain the one-way relationship between digital transformation

and economic development without paying attention to the two-way influence mechanism between digital transformation and economic development, ignoring the heterogeneity of regional development and the spatial correlation between regions. Digital transformation is achieved through digital technology. Therefore, against the background of sustainable development, it is important to study the sustainable and coordinated development of digital transformation and economic growth. In order to deeply understand the coordination relationship between digital transformation and economic growth, this paper chooses 2010 to 2019 as the research period, takes 30 provinces and cities in China (excluding Tibet) as the research objects, constructs a comprehensive evaluation index system of digital transformation and economic growth, calculates the development level and economic growth level of digital transformation in each province by the entropy method, and analyzes the coordination level and evolution relationship between digital transformation and economic growth by using the coupling coordination model and spatial autocorrelation model, thus putting forward targeted suggestions.

## 2. Materials and Methods

### 2.1. Research Methods

2.1.1. Entropy Method

In information theory, entropy is used to measure the uncertainty of things, according to which entropy can be used to judge the randomness or disorder of events. When applied to comprehensive evaluation, this can indicate the degree of dispersion of indicators, and the greater the degree of dispersion, the more important the indicators are. As an objective weighting method, in this paper, the entropy method is used to calculate the comprehensive level of digital transformation and economic development. As an objective weighting method, the entropy method can effectively avoid subjective influence [32]. The calculation steps are as follows:

1. Data standardization, with the forward and backward processing of each index, for which the equation is

$$
\begin{aligned}
\text{Positive indicators} &: r_{ij} = \frac{x_{ij} - min(x_j)}{max(x_j) - min(x_j)}, \\
\text{Negative indicators} &: r_{ij} = \frac{max(x_j) - x_{ij}}{max(x_j) - min(x_j)},
\end{aligned}
\tag{1}
$$

where $x_{ij}$ represents the $j$-th index value of the $i$-th province, and $r_{ij}$ represents the value after standardization.

2. Determine the specific gravity $P_{ij}$ of each index and calculate the index entropy $e_j$, shown as Equations (2) and (3).

$$
P_{ij} = \frac{r_{ij}}{\sum_{i=1}^{n} r_{ij}} (j = 1, 2, \cdots, m),
\tag{2}
$$

$$
e_j = -1/\ln m \times \sum_{i=1}^{n} P_{ij} ln(P_{ij}).
\tag{3}
$$

3. Calculate the information redundancy $g_j$ of the $j$-th variable and determine the weight $W_j$ of each evaluation index:

$$
g_j = 1 - e_j,
\tag{4}
$$

$$
W_j = \frac{g_j}{\sum_{j=1}^{m} g_j}.
\tag{5}
$$

4. Calculate the comprehensive score:

$$
F_i = \sum_{j=1}^{m} W_j r_{ij}.
\tag{6}
$$

### 2.1.2. Coupling Coordination Model

Coupling refers to the interdependence and mutual influence between two systems. This paper analyzes the relationship between digitalization and economic development through the coupling coordination model and explores the coordinated development level between the two systems.

The coupling degree calculation equation [33,34] is

$$C^t = \sqrt{U_1^t \times U_2^t / \left(\frac{U_1^t + U_2^t}{2}\right)^2}. \tag{7}$$

The equation for calculating the coordinated development level is

$$T^t = 0.5U_1^t + 0.5U_2^t. \tag{8}$$

The equation for calculating the coordination degree is

$$D^t = \sqrt{C^t \times T^t}. \tag{9}$$

where $U_1^t$ and $U_2^t$ represent the level of digital development and economic development, respectively. $C^t$ is the coupling degree, and $T^t$ is the comprehensive index of the coordinated development between the digital transformation level and economic development level. $D^t$ is the coupling coordination degree. Referring to the division of the coupling coordination degree in previous literature, this paper divides the coupling coordination degree into the following 10 levels (Table 1).

**Table 1.** Evaluation criteria of coupling coordination degree between digital industry and the regional economy.

| Coupling Coordination Degree | Coordination Type | Coupling Coordination Degree | Coordination Type |
|---|---|---|---|
| (0.0, 0.1) | Extreme maladjustment | [0.5, 0.6) | Grudging coordination |
| [0.1, 0.2) | Serious maladjustment | [0.6, 0.7) | Primary coordination |
| [0.2, 0.3) | Moderate maladjustment | [0.7, 0.8) | Intermediate coordination |
| [0.3, 0.4) | Mild maladjustment | [0.8, 0.9) | Good coordination |
| [0.4, 0.5) | On the verge of maladjustment | [0.9, 1.0] | High-quality coordination |

### 2.1.3. Spatial Autocorrelation Model

In this paper, the global *Moran's I* index is selected to measure the spatial correlation characteristics of the coupling coordination degree between digitalization and economic growth in China, and the calculation equation is as follows [35,36]:

$$Moran's\ I = \frac{\sum_{i=1}^{n}\sum_{j=1}^{n} w_{ij}(x_i - \bar{x})(x_j - \bar{x})}{S^2 \sum_{i=1}^{n}\sum_{j=1}^{n} w_{ij}}. \tag{10}$$

where $n$ is the total number of regions, and $x_i$ and $x_j$ are the attribute values of region $i$ and region $j$, respectively. $\bar{x}$ is the mean value of the x attribute value, $S^2$ is the variance of the attribute value, and $w_{ij}$ is the adjacent weight matrix. *Moran's I* ranges from $-1$ to 1. *Moran's I* $> 0$ indicates positive spatial correlation, and the larger its value is, the more obvious it is. *Moran's I* $< 0$ indicates negative spatial correlation, and the smaller its value is, the greater the spatial difference is. Otherwise, when *Moran's I* $= 0$, the space is random or has no spatial autocorrelation.

Then, the local *Moran's I* index is used to measure the specific location of the agglomeration effect (Equation (11)). If $I_i$ is positive, it indicates that there is similar agglomeration in region *i*. If $I_i$ is negative, there is a different agglomeration.

$$I_i = \frac{x_i - \overline{x}}{S^2} \sum_{j=1, j \neq i}^{n} w_{ij}(x_j - \overline{x}). \tag{11}$$

### 2.2. Construction of the Evaluation Index System

According to the representativeness of the selected evaluation index and the availability of data, we construct the digital transformation level index system from five dimensions— infrastructure construction, digital investment, digital benefits, digital innovation, and the information society development level—with 17 s-level indexes in total. In addition, the economic development index is measured from three aspects— economic benefits, economic structure, and social development—with seven second-level indexes in total. The evaluation index system of the digital transformation level and economic development level and its weight value is shown in Table 2.

### 2.3. Data Sources

Due to the limited availability of data, this paper selects 30 provinces and cities except Tibet as study areas. The data of 14 indexes ($u_1 \sim u_5, u_8 \sim u_{11}, u_{18} \sim u_{24}$) come from the official website of the National Bureau of Statistics (http://www.stats.gov.cn/, accessed on 11 December 2021). $u_6$ come from the China Software and Information Service Industry Development Report (https://data.cnki.net/trade/yearbook/single/n2018060071?z=z031, accessed on 11 December 2021). $u_7$ comes from the China Science and Technology Statistical Yearbook (https://data.cnki.net/trade/Yearbook/Single/N2011120098?z=Z018, accessed on 11 December 2021). The data of $u_{12}$ and $u_{13}$ are from the China Torch Statistical Yearbook (https://data.cnki.net/trade/Yearbook/Single/N2013020012?z=Z018, accessed on 11 December 2021). Four indexes ($u_{14} \sim u_{17}$) come from the China Information Society Development Report (http://www.199it.com/archives/669096.html, accessed on 11 December 2021). For some missing values, the trend value is used to compensate.

### 2.4. Descriptive Statistical Analysis

Firstly, we analyze the basic information of the data for 24 evaluation indexes from 30 provinces. The mean, standard deviation, and maximum and minimum values of each index are shown in Table 3. According to the statistical results of each evaluation index, it can be concluded that there are obvious developmental discrepancies among provinces.

**Table 2.** Index system of digital development and economic development.

| Target Layer | First-Level Index | Second-Level Index | Weight |
|---|---|---|---|
| Digital transformation ($U_1$) | Infrastructure construction ($U_{11}$) | Length of optical cable line (km) ($u_1$) | 0.0469 |
| | | Internet penetration rate (%) ($u_2$) | 0.0165 |
| | | Number of Internet access users (10,000 persons) ($u_3$) | 0.0471 |
| | Digital investment ($U_{12}$) | Number of electronic websites per 100 enterprises (unit) ($u_4$) | 0.0058 |
| | | Number of enterprises with e−commerce transaction activities (unit) ($u_5$) | 0.0807 |
| | | Investment in software business (10,000 CNY) ($u_6$) | 0.0605 |
| | | Technical transformation expenditure (10,000 CNY) ($u_7$) | 0.0523 |
| | Digital benefits ($U_{13}$) | Product development efficiency (%) ($u_8$) | 0.0849 |
| | | Software business income (10,000 CNY) ($u_9$) | 0.1169 |
| | | Information technology service income (10,000 CNY) ($u_{10}$) | 0.1139 |
| | Digital innovation ($U_{14}$) | Number of valid invention patents (unit) ($u_{11}$) | 0.1103 |
| | | Internal expenditure of R&D funds (1000 CNY) ($u_{12}$) | 0.0899 |
| | | Number of scientific and technical personnel activities (person) ($u_{13}$) | 0.0818 |
| | Information society development level ($U_{15}$) | information economy ($u_{14}$) | 0.0370 |
| | | network society ($u_{15}$) | 0.0226 |
| | | Online government | 0.0134 |
| | | Digital life ($u_{17}$) | 0.0195 |
| Economic development ($U_2$) | Economic benefits ($U_{21}$) | GDP (100 million CNY) ($u_{18}$) | 0.1772 |
| | | Total retail sales of social consumer goods (100 million CNY) ($u_{19}$) | 0.1897 |
| | | Public financial revenue (100 million yuan) ($u_{20}$) | 0.1993 |
| | Economic structure ($U_{22}$) | The ratio of secondary and tertiary industries to GDP (%) ($u_{21}$) | 0.0391 |
| | | Per capita disposable income of residents (CNY) ($u_{22}$) | 0.1200 |
| | Social development ($U_{23}$) | Average wage of on−the−job workers in urban units (CNY) ($u_{23}$) | 0.1324 |
| | | Number of permanent residents at the end of the year (10,000 persons) ($u_{24}$) | 0.1423 |



**Table 3.** Descriptive statistical analysis.

| Variables | Obs | Mean | Std. Dev. | Min | Max |
|---|---|---|---|---|---|
| $u_1$ | 300 | 858,931.30 | 721,418.50 | 41,531.40 | 3,679,239.00 |
| $u_2$ | 300 | 49.98 | 13.92 | 19.80 | 84.20 |
| $u_3$ | 300 | 867.91 | 747.93 | 34.71 | 3801.60 |
| $u_4$ | 300 | 53.07 | 10.33 | 15.00 | 80.58 |
| $u_5$ | 300 | 2328.78 | 2892.30 | 15.21 | 15,175.00 |
| $u_6$ | 300 | 549.80 | 587.67 | 3.17 | 3439.83 |
| $u_7$ | 300 | 1,196,976.00 | 1,178,563.00 | 13,134.00 | 7,178,935.00 |
| $u_8$ | 300 | 50,000,000.00 | 69,200,000.00 | 52,338.43 | 430,000,000.00 |
| $u_9$ | 300 | 13,800,000.00 | 22,400,000.00 | 1855.16 | 120,000,000.00 |
| $u_{10}$ | 300 | 7,508,418.00 | 12,600,000.00 | 3647.90 | 79,500,000.00 |
| $u_{11}$ | 300 | 20,044.78 | 42,858.93 | 69.34 | 375,515.00 |
| $u_{12}$ | 300 | 21,800,000.00 | 35,300,000.00 | 72,678.00 | 289,000,000.00 |
| $u_{13}$ | 300 | 169,101.20 | 234,636.80 | 1562.00 | 1,630,471.00 |
| $u_{14}$ | 300 | 0.37 | 0.11 | 0.23 | 0.83 |
| $u_{15}$ | 300 | 0.40 | 0.10 | 0.22 | 0.73 |
| $u_{16}$ | 300 | 0.61 | 0.19 | 0.15 | 1.07 |
| $u_{17}$ | 300 | 0.48 | 0.19 | 0.11 | 0.99 |
| $u_{18}$ | 300 | 23,346.53 | 19,188.70 | 1144.20 | 107,986.90 |
| $u_{19}$ | 300 | 9505.75 | 8070.19 | 351.00 | 42,951.80 |
| $u_{20}$ | 300 | 1940.65 | 1756.61 | 88.94 | 10,063.95 |
| $u_{21}$ | 300 | 0.90 | 0.05 | 0.74 | 1.00 |
| $u_{22}$ | 300 | 21,822.51 | 10,429.57 | 7203.54 | 69,442.00 |
| $u_{23}$ | 300 | 60,771.88 | 22,578.23 | 29,092.00 | 173,205.00 |
| $u_{24}$ | 300 | 4557.63 | 2759.50 | 563.00 | 12,489.00 |

## 2.5. Correlation Analysis

Firstly, this study analyzes the correlation between digital transformation and economic development variables and uses SPSS 23.0 to perform a Pearson correlation analysis on the data. The calculation principle is as shown in Formula 12, where $r$ represents the correlation coefficient, with a value between $-1$ and 1, and $X$ and $Y$ represent two variables, respectively. The closer the absolute value of $r$ is to 1, the stronger the correlation between two variables, and the closer it is to 0, the weaker the correlation between variables. The analysis results are shown in Table 4.

$$r = \frac{\sum_{i=1}^{n}\left(X_i - \overline{X}\right)\left(Y_i - \overline{Y}\right)}{\sqrt{\sum_{i=1}^{n}\left(X_i - \overline{X}\right)^2}\sqrt{\sum_{i=1}^{n}\left(Y_i - \overline{Y}\right)^2}} \tag{12}$$

**Table 4.** Pearson correlation analysis.

| Pearson | $u_{18}$ | $u_{19}$ | $u_{20}$ | $u_{21}$ | $u_{22}$ | $u_{23}$ | $u_{24}$ |
|---|---|---|---|---|---|---|---|
| $u_1$ | 0.834 ** | 0.782 ** | 0.669 ** | 0.223 | 0.067 | −0.149 | 0.794 ** |
| $u_2$ | 0.391 * | 0.403 * | 0.665 ** | 0.699 ** | 0.905 ** | 0.811 ** | −0.108 |
| $u_3$ | 0.970 ** | 0.980 ** | 0.871 ** | 0.376 * | 0.358 | 0.127 | 0.831 ** |
| $u_4$ | 0.334 | 0.229 | 0.448* | 0.250 | 0.465 ** | 0.413 * | −0.013 |
| $u_5$ | 0.770 ** | 0.746 ** | 0.787 ** | 0.464 ** | 0.487 ** | 0.268 | 0.450* |
| $u_6$ | 0.792 ** | 0.733 ** | 0.706 ** | 0.309 | 0.263 | 0.122 | 0.640 ** |
| $u_7$ | 0.799 ** | 0.788 ** | 0.690 ** | 0.308 | 0.282 | 0.100 | 0.696 ** |
| $u_8$ | 0.917 ** | 0.901 ** | 0.911 ** | 0.538 ** | 0.540 ** | 0.369 * | 0.591 ** |
| $u_9$ | 0.753 ** | 0.753 ** | 0.893 ** | 0.560 ** | 0.651 ** | 0.508 ** | 0.387 * |
| $u_{10}$ | 0.527 ** | 0.575 ** | 0.703 ** | 0.481 ** | 0.615 ** | 0.468 ** | 0.250 |
| $u_{11}$ | 0.827 ** | 0.805 ** | 0.837 ** | 0.433* | 0.383* | 0.266 | 0.559 ** |
| $u_{12}$ | 0.701 ** | 0.659 ** | 0.861 ** | 0.656 ** | 0.815 ** | 0.728 ** | 0.322 |
| $u_{13}$ | 0.784 ** | 0.753 ** | 0.922 ** | 0.674 ** | 0.800 ** | 0.687 ** | 0.414 * |
| $u_{14}$ | 0.224 | 0.230 | 0.561 ** | 0.707 ** | 0.937 ** | 0.950 ** | −0.199 |
| $u_{15}$ | 0.333 | 0.353 | 0.619 ** | 0.594 ** | 0.931 ** | 0.845 ** | −0.129 |
| $u_{16}$ | 0.327 | 0.311 | 0.524 ** | 0.376* | 0.710 ** | 0.587 ** | 0.079 |
| $u_{17}$ | 0.376 * | 0.382 * | 0.678 ** | 0.727 ** | 0.956 ** | 0.894 ** | −0.102 |

** At the 0.01 level (double tail), the correlation is significant. * At the 0.05 level (double tail), the correlation is significant.

The Pearson correlation coefficient results in Table 4 show that there is an obvious correlation between digital transformation and economic development variables, which indicates that the index selection is reasonable and suitable for coupling coordination degree analysis. Notably, through Pearson correlation coefficient table analysis, it can be found that the correlation between the infrastructure construction, digital investment, digital benefits, and economic benefits of digital innovation is obvious. The correlation between innovation and the information society development level and economic structure is also remarkable.

## 3. Results

### 3.1. Digital Transformation Level and Economic Development Level Result Analysis

From 2010 to 2019, the level of China's digital transformation was in a fast-rising stage (Table 5). In 2010 and 2019, the average development value of China's digital transformation was 0.076 and 0.243, respectively. The overall level of digital transformation improved significantly, indicating that China's digital transformation was in a fast-rising stage. Among all provinces in China, only Liaoning Province showed a comprehensive numerical downward trend in 2016. Combined with the analysis of specific indexes, the main reason lies in the decline of software business investment, technical transformation expenditure, software business income, and information technology service income in Liaoning Province in 2016, which means that the digital investment level and digital income effect level of Liaoning Province in 2016 were both low. The average value represents the overall level of digital transformation and development in each region for five years. From the calculation results of the average value, Guangdong Province has the highest digital level index, which is 0.514, while Qinghai's digital development water average value is 0.038. The difference between them is nearly 13-fold, indicating that with the acceleration of the digitalization process in various provinces and cities, there is a big gap between provinces. The top 10 provinces with the highest level of digital development are Guangdong, Jiangsu, Zhejiang, Beijing, Shandong, Shanghai, Fujian, Sichuan, Liaoning, and Hubei. Except Sichuan and Liaoning, they are all in the east, indicating that the digitalization level of the eastern provinces is generally high, and they are the most important areas for China's economic development. Regions ranking from 11 to 20 are mainly in the central region of China, with two western provinces (Shaanxi and Chongqing), and their digital transformation and development level is medium. The provinces ranking 21 to 30 are in northeast China (Jilin, Heilongjiang) and west China, and the comprehensive index of the digital development level in these provinces is low. From this ranking, it can be concluded that the development level of China's digital transformation is characterized by an agglomeration in space, and the development level of the agglomeration areas in the eastern region is higher, followed by the central and northeastern regions, with the lowest level in the western region.

It can be seen from Table 6 that China's economic development showed a gradual upward trend from 2010 to 2019, and the overall level of economic growth in 2010 and 2019 was 0.152 and 0.337, respectively, indicating that the level of economic growth in those 10 years obviously improved. In 2010, there were few provinces with a comprehensive index of economic growth level higher than 0.5—only Guangdong Province and Jiangsu Province. The economic growth level in the central and northeastern regions was relatively low, and was in the stage of low-speed economic growth. The average economic development index in the western region was below 0.2, which was the lowest level in China. By 2019, with the rapid development of China's economy, six provinces had an economic growth level above 0.5: Guangdong, Jiangsu, Shandong, Zhejiang, Shanghai, and Beijing. On average, the top 10 provinces with the highest level of economic development in China are in the eastern region (Guangdong, Jiangsu, Shandong, Zhejiang, Shanghai, Beijing and Hebei), with two in central regions (Henan and Hubei) and one western province—Sichuan. The provinces ranked from 10 to 20 are in the eastern region (Fujian and Tianjin), the central region (Hunan, Anhui, Jiangxi and Shanxi), the northeast region (Liaoning) and the western

region (Chongqing, Shaanxi and Yunnan). The provinces ranked from 21 to 30 are from northeast China (Jilin and Heilongjiang) and other western provinces. From the perspective of spatial patterns, China's economic development level is also characterized as "high in the east and low in the west".

Combined with the comprehensive index of digital transformation and economic growth and development, it can be concluded that the areas with high levels of digital transformation and economic growth are distributed in the eastern region. The eastern region is ahead of other regions in China in terms of geographical location, economic foundation, infrastructure construction, scientific and technological innovation, and human resources. On the contrary, the western provinces have a poor geographical position and relatively backward economy, and their infrastructure construction and scientific and technological innovation levels are also low from the perspective of the whole country, so they are areas with low levels of digital transformation and economic growth in China.

**Table 5.** Comprehensive index of digital development level from 2010 to 2019.

| Province | 2010 | 2011 | 2012 | 2013 | 2014 | 2015 | 2016 | 2017 | 2018 | 2019 | Average |
|---|---|---|---|---|---|---|---|---|---|---|---|
| Guangdong | 0.219 | 0.251 | 0.289 | 0.372 | 0.425 | 0.504 | 0.607 | 0.716 | 0.827 | 0.928 | 0.514 |
| Jiangsu | 0.246 | 0.296 | 0.340 | 0.381 | 0.448 | 0.500 | 0.558 | 0.583 | 0.642 | 0.685 | 0.468 |
| Zhejiang | 0.178 | 0.200 | 0.218 | 0.255 | 0.296 | 0.339 | 0.377 | 0.412 | 0.465 | 0.507 | 0.325 |
| Beijing | 0.172 | 0.191 | 0.213 | 0.235 | 0.269 | 0.302 | 0.332 | 0.381 | 0.440 | 0.512 | 0.305 |
| Shandong | 0.138 | 0.156 | 0.181 | 0.221 | 0.245 | 0.274 | 0.320 | 0.344 | 0.389 | 0.407 | 0.268 |
| Shanghai | 0.167 | 0.179 | 0.196 | 0.216 | 0.245 | 0.267 | 0.288 | 0.308 | 0.333 | 0.368 | 0.257 |
| Fujian | 0.080 | 0.099 | 0.111 | 0.130 | 0.151 | 0.176 | 0.211 | 0.228 | 0.255 | 0.277 | 0.172 |
| Sichuan | 0.084 | 0.099 | 0.106 | 0.131 | 0.150 | 0.174 | 0.203 | 0.228 | 0.252 | 0.288 | 0.171 |
| Liaoning | 0.123 | 0.129 | 0.126 | 0.147 | 0.167 | 0.170 | 0.151 | 0.156 | 0.151 | 0.173 | 0.149 |
| Hubei | 0.063 | 0.071 | 0.084 | 0.112 | 0.134 | 0.156 | 0.180 | 0.196 | 0.224 | 0.252 | 0.147 |
| Anhui | 0.056 | 0.076 | 0.086 | 0.101 | 0.124 | 0.146 | 0.177 | 0.197 | 0.226 | 0.252 | 0.144 |
| Hunan | 0.070 | 0.083 | 0.099 | 0.115 | 0.128 | 0.148 | 0.165 | 0.189 | 0.201 | 0.223 | 0.142 |
| Tianjin | 0.083 | 0.091 | 0.103 | 0.116 | 0.128 | 0.144 | 0.156 | 0.162 | 0.181 | 0.202 | 0.137 |
| Henan | 0.063 | 0.076 | 0.083 | 0.103 | 0.118 | 0.137 | 0.160 | 0.171 | 0.190 | 0.204 | 0.131 |
| Hebei | 0.057 | 0.070 | 0.078 | 0.090 | 0.102 | 0.114 | 0.131 | 0.152 | 0.186 | 0.209 | 0.119 |
| Shaanxi | 0.050 | 0.058 | 0.066 | 0.078 | 0.095 | 0.117 | 0.137 | 0.152 | 0.177 | 0.207 | 0.114 |
| Chongqing | 0.048 | 0.059 | 0.067 | 0.080 | 0.097 | 0.114 | 0.136 | 0.147 | 0.161 | 0.182 | 0.109 |
| Jiangxi | 0.033 | 0.042 | 0.049 | 0.067 | 0.074 | 0.087 | 0.100 | 0.137 | 0.167 | 0.196 | 0.095 |
| Guangxi | 0.036 | 0.045 | 0.051 | 0.059 | 0.066 | 0.074 | 0.085 | 0.098 | 0.114 | 0.148 | 0.078 |
| Shanxi | 0.047 | 0.055 | 0.061 | 0.064 | 0.071 | 0.077 | 0.081 | 0.090 | 0.099 | 0.111 | 0.076 |
| Jilin | 0.038 | 0.048 | 0.050 | 0.054 | 0.069 | 0.073 | 0.080 | 0.087 | 0.100 | 0.134 | 0.073 |
| Inner Mongolia | 0.036 | 0.041 | 0.045 | 0.052 | 0.064 | 0.066 | 0.073 | 0.087 | 0.095 | 0.114 | 0.067 |
| Heilongjiang | 0.038 | 0.043 | 0.046 | 0.053 | 0.063 | 0.067 | 0.073 | 0.083 | 0.087 | 0.097 | 0.065 |
| Yunnan | 0.025 | 0.032 | 0.035 | 0.043 | 0.052 | 0.064 | 0.075 | 0.082 | 0.098 | 0.117 | 0.062 |
| Guizhou | 0.017 | 0.026 | 0.034 | 0.041 | 0.050 | 0.058 | 0.070 | 0.079 | 0.094 | 0.112 | 0.058 |
| Xinjiang | 0.026 | 0.031 | 0.034 | 0.042 | 0.050 | 0.059 | 0.062 | 0.066 | 0.080 | 0.085 | 0.054 |
| Hainan | 0.029 | 0.033 | 0.037 | 0.043 | 0.050 | 0.054 | 0.058 | 0.063 | 0.071 | 0.079 | 0.052 |
| Gansu | 0.017 | 0.023 | 0.029 | 0.037 | 0.045 | 0.049 | 0.056 | 0.062 | 0.073 | 0.084 | 0.047 |
| Ningxia | 0.018 | 0.026 | 0.026 | 0.032 | 0.039 | 0.042 | 0.044 | 0.049 | 0.057 | 0.065 | 0.040 |
| Qinghai | 0.015 | 0.019 | 0.024 | 0.031 | 0.036 | 0.042 | 0.045 | 0.048 | 0.055 | 0.065 | 0.038 |
| Overall level | 0.076 | 0.088 | 0.099 | 0.117 | 0.135 | 0.153 | 0.173 | 0.192 | 0.216 | 0.243 | 0.149 |

**Table 6.** Comprehensive index of economic development level from 2010 to 2019.

| Province | 2010 | 2011 | 2012 | 2013 | 2014 | 2015 | 2016 | 2017 | 2018 | 2019 | Average |
|---|---|---|---|---|---|---|---|---|---|---|---|
| Guangdong | 0.406 | 0.457 | 0.496 | 0.534 | 0.582 | 0.633 | 0.688 | 0.747 | 0.817 | 0.869 | 0.623 |
| Jiangsu | 0.342 | 0.391 | 0.432 | 0.474 | 0.515 | 0.556 | 0.590 | 0.630 | 0.692 | 0.730 | 0.535 |
| Shandong | 0.310 | 0.347 | 0.381 | 0.431 | 0.468 | 0.500 | 0.533 | 0.569 | 0.552 | 0.579 | 0.467 |
| Zhejiang | 0.268 | 0.304 | 0.332 | 0.360 | 0.389 | 0.420 | 0.457 | 0.499 | 0.548 | 0.591 | 0.417 |
| Shanghai | 0.254 | 0.282 | 0.302 | 0.330 | 0.361 | 0.395 | 0.439 | 0.470 | 0.523 | 0.547 | 0.390 |
| Beijing | 0.229 | 0.264 | 0.291 | 0.319 | 0.348 | 0.379 | 0.407 | 0.438 | 0.489 | 0.525 | 0.369 |
| Henan | 0.221 | 0.246 | 0.268 | 0.288 | 0.311 | 0.332 | 0.355 | 0.385 | 0.423 | 0.451 | 0.328 |
| Sichuan | 0.196 | 0.223 | 0.247 | 0.272 | 0.294 | 0.314 | 0.335 | 0.362 | 0.401 | 0.428 | 0.307 |
| Hubei | 0.164 | 0.189 | 0.211 | 0.234 | 0.260 | 0.284 | 0.306 | 0.332 | 0.375 | 0.401 | 0.276 |
| Hebei | 0.186 | 0.210 | 0.228 | 0.251 | 0.269 | 0.287 | 0.307 | 0.334 | 0.329 | 0.348 | 0.275 |
| Hunan | 0.163 | 0.186 | 0.206 | 0.230 | 0.251 | 0.272 | 0.293 | 0.318 | 0.336 | 0.358 | 0.261 |
| Fujian | 0.147 | 0.172 | 0.194 | 0.214 | 0.235 | 0.253 | 0.272 | 0.297 | 0.345 | 0.372 | 0.250 |
| Anhui | 0.152 | 0.175 | 0.194 | 0.211 | 0.228 | 0.245 | 0.263 | 0.287 | 0.337 | 0.359 | 0.245 |
| Liaoning | 0.172 | 0.196 | 0.217 | 0.249 | 0.260 | 0.258 | 0.258 | 0.273 | 0.265 | 0.278 | 0.243 |
| Tianjin | 0.128 | 0.143 | 0.162 | 0.186 | 0.203 | 0.221 | 0.237 | 0.252 | 0.247 | 0.261 | 0.204 |
| Shaanxi | 0.115 | 0.132 | 0.150 | 0.166 | 0.179 | 0.189 | 0.201 | 0.225 | 0.252 | 0.270 | 0.188 |
| Jiangxi | 0.111 | 0.129 | 0.145 | 0.162 | 0.178 | 0.194 | 0.208 | 0.225 | 0.251 | 0.269 | 0.187 |
| Chongqing | 0.105 | 0.125 | 0.140 | 0.156 | 0.175 | 0.192 | 0.208 | 0.225 | 0.256 | 0.275 | 0.186 |
| Yunnan | 0.105 | 0.119 | 0.133 | 0.148 | 0.158 | 0.172 | 0.187 | 0.208 | 0.241 | 0.263 | 0.173 |
| Shanxi | 0.118 | 0.136 | 0.150 | 0.161 | 0.168 | 0.174 | 0.180 | 0.204 | 0.215 | 0.229 | 0.173 |
| Guangxi | 0.103 | 0.114 | 0.128 | 0.146 | 0.161 | 0.177 | 0.191 | 0.207 | 0.221 | 0.235 | 0.168 |
| Inner Mongolia | 0.100 | 0.117 | 0.132 | 0.154 | 0.166 | 0.177 | 0.189 | 0.197 | 0.200 | 0.216 | 0.165 |
| Heilongjiang | 0.104 | 0.116 | 0.125 | 0.140 | 0.150 | 0.156 | 0.165 | 0.176 | 0.159 | 0.169 | 0.146 |
| Jilin | 0.084 | 0.099 | 0.114 | 0.134 | 0.146 | 0.155 | 0.169 | 0.180 | 0.164 | 0.171 | 0.142 |
| Guizhou | 0.075 | 0.090 | 0.104 | 0.118 | 0.130 | 0.144 | 0.157 | 0.173 | 0.205 | 0.217 | 0.141 |
| Xinjiang | 0.057 | 0.077 | 0.089 | 0.102 | 0.114 | 0.125 | 0.133 | 0.147 | 0.166 | 0.176 | 0.119 |
| Gansu | 0.058 | 0.068 | 0.079 | 0.092 | 0.103 | 0.112 | 0.120 | 0.130 | 0.142 | 0.149 | 0.105 |
| Ningxia | 0.045 | 0.055 | 0.063 | 0.069 | 0.077 | 0.085 | 0.094 | 0.105 | 0.119 | 0.130 | 0.084 |
| Qinghai | 0.036 | 0.044 | 0.050 | 0.058 | 0.067 | 0.077 | 0.086 | 0.098 | 0.112 | 0.121 | 0.075 |
| Hainan | 0.021 | 0.031 | 0.041 | 0.053 | 0.064 | 0.076 | 0.084 | 0.098 | 0.116 | 0.128 | 0.071 |
| Overall level | 0.152 | 0.175 | 0.194 | 0.215 | 0.234 | 0.252 | 0.270 | 0.293 | 0.317 | 0.337 | 0.244 |

### 3.2. Analysis of Coupling Coordination Results

According to the coupling coordination model, the coupling coordination results of digital level and economic growth in China from 2010 to 2019 are calculated (Table 7). Referring to the classification of coupling coordination types (Table 1), the obtained coupling coordination degree involves nine coordination types, ranging from severe imbalance to high-quality coordination.

From the time series, from 2010 to 2019, the coupling coordination degree of digital transformation and economic development in 30 provinces and cities in China increased year by year, from 0.307 in 2010 to 0.509 in 2019, which indicates that the mutual influence between digital transformation level and economic growth increased, and the coordination degree between the two systems was increasing. In 2010, only Guangdong and Jiangsu were barely coordinated, and other provinces were out of balance. In 2019, Guangdong achieved high-quality coordination, Jiangsu achieved good coordination, Zhejiang and Beijing achieved intermediate coordination, Shandong and Shanghai achieved primary coordination, Sichuan, Fujian, Hubei, Henan, Anhui, Hunan, and Hebei were barely coordinated, and the remaining 17 provinces were out of balance. Compared with 2013, the degree of coupled coordination was significantly improved.

**Table 7.** Coupling and coordination results of digital level and economic development from 2010 to 2019.

| Province | 2010 | 2011 | 2012 | 2013 | 2014 | 2015 | 2016 | 2017 | 2018 | 2019 | Average |
|---|---|---|---|---|---|---|---|---|---|---|---|
| Guangdong | 0.546 | 0.582 | 0.616 | 0.667 | 0.705 | 0.752 | 0.804 | 0.855 | 0.907 | 0.948 | 0.738 |
| Jiangsu | 0.538 | 0.583 | 0.619 | 0.652 | 0.693 | 0.726 | 0.757 | 0.778 | 0.816 | 0.841 | 0.700 |
| Zhejiang | 0.467 | 0.496 | 0.519 | 0.551 | 0.583 | 0.614 | 0.645 | 0.673 | 0.711 | 0.740 | 0.600 |
| Shandong | 0.455 | 0.482 | 0.512 | 0.555 | 0.582 | 0.609 | 0.643 | 0.665 | 0.681 | 0.697 | 0.588 |
| Beijing | 0.445 | 0.474 | 0.499 | 0.523 | 0.553 | 0.582 | 0.606 | 0.639 | 0.681 | 0.720 | 0.572 |
| Shanghai | 0.454 | 0.474 | 0.493 | 0.517 | 0.545 | 0.570 | 0.596 | 0.617 | 0.646 | 0.670 | 0.558 |
| Sichuan | 0.359 | 0.385 | 0.402 | 0.434 | 0.458 | 0.483 | 0.511 | 0.536 | 0.564 | 0.592 | 0.472 |
| Henan | 0.344 | 0.370 | 0.386 | 0.415 | 0.437 | 0.462 | 0.489 | 0.507 | 0.533 | 0.551 | 0.449 |
| Fujian | 0.330 | 0.361 | 0.383 | 0.408 | 0.434 | 0.460 | 0.490 | 0.510 | 0.545 | 0.567 | 0.449 |
| Hubei | 0.319 | 0.341 | 0.365 | 0.402 | 0.432 | 0.459 | 0.485 | 0.505 | 0.538 | 0.564 | 0.441 |
| Liaoning | 0.382 | 0.399 | 0.407 | 0.438 | 0.456 | 0.457 | 0.445 | 0.454 | 0.447 | 0.468 | 0.435 |
| Hunan | 0.327 | 0.352 | 0.378 | 0.404 | 0.423 | 0.448 | 0.469 | 0.495 | 0.510 | 0.531 | 0.434 |
| Anhui | 0.303 | 0.339 | 0.359 | 0.382 | 0.410 | 0.435 | 0.464 | 0.487 | 0.525 | 0.548 | 0.425 |
| Hebei | 0.321 | 0.348 | 0.366 | 0.388 | 0.407 | 0.426 | 0.448 | 0.475 | 0.497 | 0.519 | 0.419 |
| Tianjin | 0.320 | 0.338 | 0.360 | 0.383 | 0.402 | 0.422 | 0.438 | 0.450 | 0.460 | 0.479 | 0.405 |
| Shaanxi | 0.275 | 0.295 | 0.315 | 0.337 | 0.361 | 0.386 | 0.407 | 0.430 | 0.460 | 0.487 | 0.375 |
| Chongqing | 0.267 | 0.293 | 0.311 | 0.335 | 0.361 | 0.385 | 0.410 | 0.427 | 0.451 | 0.473 | 0.371 |
| Jiangxi | 0.245 | 0.272 | 0.291 | 0.322 | 0.339 | 0.360 | 0.380 | 0.420 | 0.452 | 0.479 | 0.356 |
| Shanxi | 0.273 | 0.293 | 0.308 | 0.319 | 0.330 | 0.340 | 0.348 | 0.368 | 0.382 | 0.399 | 0.336 |
| Guangxi | 0.247 | 0.268 | 0.284 | 0.305 | 0.321 | 0.338 | 0.357 | 0.377 | 0.398 | 0.432 | 0.333 |
| Inner Mongolia | 0.245 | 0.263 | 0.277 | 0.299 | 0.320 | 0.329 | 0.343 | 0.361 | 0.372 | 0.397 | 0.321 |
| Yunnan | 0.226 | 0.248 | 0.261 | 0.282 | 0.302 | 0.323 | 0.345 | 0.362 | 0.392 | 0.419 | 0.316 |
| Jilin | 0.237 | 0.262 | 0.275 | 0.291 | 0.317 | 0.326 | 0.341 | 0.354 | 0.358 | 0.389 | 0.315 |
| Heilongjiang | 0.251 | 0.266 | 0.275 | 0.294 | 0.311 | 0.320 | 0.331 | 0.348 | 0.343 | 0.358 | 0.310 |
| Guizhou | 0.190 | 0.219 | 0.244 | 0.264 | 0.284 | 0.302 | 0.324 | 0.341 | 0.373 | 0.395 | 0.294 |
| Xinjiang | 0.197 | 0.221 | 0.235 | 0.256 | 0.275 | 0.293 | 0.301 | 0.314 | 0.340 | 0.350 | 0.278 |
| Gansu | 0.178 | 0.198 | 0.219 | 0.241 | 0.260 | 0.272 | 0.286 | 0.300 | 0.319 | 0.335 | 0.261 |
| Hainan | 0.157 | 0.179 | 0.196 | 0.218 | 0.237 | 0.254 | 0.264 | 0.280 | 0.302 | 0.317 | 0.240 |
| Ningxia | 0.170 | 0.195 | 0.201 | 0.216 | 0.234 | 0.244 | 0.255 | 0.268 | 0.287 | 0.303 | 0.237 |
| Qinghai | 0.153 | 0.171 | 0.187 | 0.205 | 0.222 | 0.238 | 0.250 | 0.262 | 0.280 | 0.297 | 0.227 |
| Overall level | 0.307 | 0.332 | 0.351 | 0.377 | 0.400 | 0.420 | 0.441 | 0.462 | 0.486 | 0.509 | 0.409 |

From 2010 to 2019, the regions with an average coupling coordination degree above 0.4 were all eastern provinces (Guangdong, Jiangsu, Zhejiang, Shandong, Beijing and Shanghai), which indicates that the digital transformation level and economic development of these provinces were in an effective coupling stage, and their digital transformation activities developed more rapidly than those of other provinces, effectively driving economic growth in all walks of life. The areas with an average degree of coordination between 0.4 and 0.5 (on the verge of disorder) included Fujian, Hebei, and Tianjin in the east, Henan, Hubei, Hunan, and Anhui in the middle, Liaoning in the northeast, and Sichuan in the west. Among them, the coupling coordination degree of Anhui Province increased rapidly, from 0.303 to 0.548 in 2010, indicating that its digital transformation level and economic development level developed rapidly. The coupling coordination degree of Liaoning Province decreased slightly in 2016, which was due to the decline of its digital transformation level and the upward trend of its economic growth level in 2016. Other provinces have developed steadily, and the level of coupling and coordination has increased by one level. Finally, the areas with an average coupling coordination degree between 0.2 and 0.4 (from moderate to near-maladjustment) include 15 provinces, comprising Hainan in the east, Jilin and Heilongjiang in the northeast, Jiangxi and Shanxi in the middle, and most areas in the west. The levels of digital transformation and economic development in these areas are not high, and they are far behind other provinces. Due to the lack of digital infrastructure, the relatively low economic level, and scientific and technological capabilities, the two systems have not yet formed a benign and interactive coordinated development.

Combined with the comparison of the development level index of the two systems obtained in Section 3.1, it can be found that all provinces in China except Guangdong exhibit digital lag coordinated development. This is because China's digital transformation depends on the development of new technologies, and most enterprises are still in the early stage of digital transformation, so the digital development speed does not match the economic growth speed, which leads to the coordinated development of digital lag on the whole. However, with the continuous development of digital transformation, the gap between them is reduced, and the degree of coupling coordination between them is improved.

### 3.3. Spatial Correlation Analysis of Digital Development and Economic Growth Coordination

In order to further analyze the spatial evolution characteristics of the coupling coordination degree between digitalization and economic growth and observe whether there is any correlation between them in space, this paper uses Stata software to calculate the global *Moran's I* index from 2010 to 2019. The calculation results are shown in Table 8.

**Table 8.** Global *Moran's I* Index Value from 2010 to 2019.

| Year | 2010 | 2011 | 2012 | 2013 | 2014 | 2015 | 2016 | 2017 | 2018 | 2019 |
|------|------|------|------|------|------|------|------|------|------|------|
| *Moran's I* | 0.238 | 0.249 | 0.256 | 0.239 | 0.237 | 0.242 | 0.244 | 0.237 | 0.248 | 0.234 |

From 2010 to 2019, the global *Moran's I* index values were all greater than 0.230, and the p values were all less than 0.05, indicating that the probability of data aggregation was greater than the probability of random distribution and allowing us to reject the original hypothesis. This indicates that there is a positive spatial correlation between the coupling coordination degree of digital transformation development and economic development level in China; that is, the provinces with a high coupling coordination degree are geographically adjacent, and the provinces with a low coupling coordination degree are also clustered together.

The value of the global autocorrelation index reveals the overall correlation characteristics of the coupling coordination degree between digital transformation and economic growth, but it cannot specifically express the agglomeration relationship between provinces and neighboring provinces. In order to know more about the spatial agglomeration of provinces and cities, we measured the specific provinces with a spatial agglomeration and the degree of association between provinces through the local spatial autocorrelation index. We used the coupling coordination degree in 2010 and 2019 to calculate the local Moran index, as shown in Figure 1. The Moran scatter plot shows that most provinces fall in the first quadrant (high–high agglomeration area) and the third quadrant (low–low agglomeration area). A high–high agglomeration area indicates that provinces with a high level of coupling coordination between digital level and economic development level are also surrounded by provinces with high coupling coordination. A low–low agglomeration area indicates that provinces with low coupling coordination degree have a lower coupling coordination degree with neighboring provinces. This further indicates that provinces with a high coupling coordination degree are more concentrated in space. In addition, the second quadrant (low–high agglomeration area) indicates that provinces with a low coupling coordination degree are surrounded by provinces with a high coupling coordination level, and the fourth quadrant (high–low agglomeration area) indicates that provinces with a high coupling coordination degree are surrounded by provinces with a low coupling coordination level. The specific agglomeration distribution of 30 provinces is shown in Table 9.

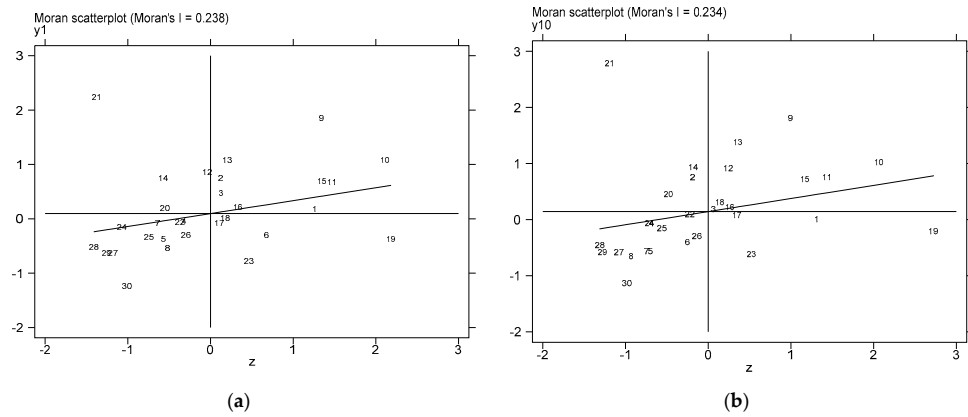

**Figure 1.** Moran scatter chart. (**a**) 2010; (**b**) 2019.

**Table 9.** Classification of agglomeration in 30 provinces and cities.

| Aggregation Area Type | 2010 | 2019 |
|---|---|---|
| High–high area | Beijing, Tianjin, Hebei, Shanghai, Jiangsu, Zhejiang, Fujian, Shandong, and Henan | Hebei, Shanghai, Jiangsu, Zhejiang, Anhui, Fujian, Shandong, Henan, and Hunan |
| Low–low area | Shanxi, Inner Mongolia Jilin, Heilongjiang, Chongqing, Guizhou, Yunnan, Shaanxi, Gansu, Qinghai, Ningxia, and Xinjiang | Shanxi, Inner Mongolia, Liaoning, Jilin, Heilongjiang, Chongqing, Guizhou, Yunnan, Shaanxi, Gansu, Qinghai, Ningxia, and Xinjiang |
| Low–high area | Anhui, Jiangxi, Guangxi, and Hainan | Tianjin, Jiangxi, Guangxi, and Hainan |
| High–low area | Liaoning, Hubei, Hunan, Guangdong, and Sichuan | Beijing, Hubei, Guangdong, and Sichuan |

1. High–high agglomeration area. From the perspective of space, these provinces have changed greatly in the studied 10 years. In 2010, these areas were mainly concentrated in Beijing–Tianjin–Hebei, Yangtze River Delta and provinces close to these two regions, with a total of nine provinces. In 2019, this evolved into the five east China provinces of Shanghai, Jiangsu, Zhejiang, Shandong, and Fujian, the two central China provinces of Henan and Hunan, and Hebei Province. This shows that the spatial agglomeration relationship between east China and central China is obviously enhanced, and its digitalization and economic development effect is remarkable. These provinces have a high coordination index of digital level and economic growth, and they are representative provinces to achieve coordinated development, among which Guangdong, Jiangsu and Zhejiang are advantageous regions with large investments in digital transformation and better economic development in China. The Yangtze River Delta and its surrounding provinces have a high level of urban development, which is supported by sufficient digital foundation and digital industry. The neighboring provinces have close economic ties and can form a good interactive relationship, with obvious spillover effects such as factor flow and technology diffusion, thus mutually driving the continuous improvement of the coupling coordination level of such regions. In the future, this type of area should continue to maintain a good development trend, give full play to its advantages in technology, economy, manpower, geographical location, and other factors, increase digital investment, and continuously improve its digital transformation degree. At the same time, the area should also give full play to its radiation role and support the transmission technology, talents, capital, etc., of provinces with low coupling coordination to promote the coordinated development of low-level provincial digitalization and economic growth.

2. Low–low agglomeration area. In 2010, there were 12 provinces in this classification, and in 2019, there were 13 provinces, which were mainly concentrated in the 3 northeastern provinces, the central part (Shanxi, Inner Mongolia), and most western provinces, where the spatial changes were relatively stable. The geographical environment in this area is poor, so the economic development is slow. The coupling

coordination index in these places is low, which greatly restricts digitalization development and economic growth and has a relatively significant negative impact in space. Due to the imperfect infrastructure, poor scientific and technological ability, single industrial structure, and weak industrial base, the growth rate of the digital level and economic development level of these provinces is also relatively slow. It is far from sufficient for such areas to rely solely on their own efforts to improve the level of digitalization and economic development. For provinces that started digitalization late, digital infrastructure construction and digital investment can be implemented, such as continuing to strengthen the construction of optical cable lines, increasing the internet penetration rate, increasing the investment of funds and talents in scientific research and development, and continuously promoting digital construction. In addition, these provinces need to rely on national policies to help them and formulate countermeasures suitable for their development. While increasing external support, they also need to learn the technical knowledge and management systems of high-level areas to break the deadlock in the development of low concentration in these areas.

3.  Low–high agglomeration area. In 2010, this category included the four provinces of Anhui, Jiangxi, Guangxi, and Hainan. By 2019, Anhui Province had become a high–high agglomeration area, which indicated that the coordination level between its own digital transformation and economic development continued to rise by 2019. However, Tianjin changed from a high–high agglomeration area to a low–high agglomeration type, which indicates that the coupling coordination degree of Tianjin is growing slowly, while the coordination degrees of neighboring provinces such as Hebei and Beijing are developing relatively fast. These provinces with a low degree of coupling and coordination are still in the middle and lower reaches in terms of economic development level and digital development. Geographically, such areas are in the transition zone from high-level areas with high coupling coordination to low-level areas. These four provinces are all close to Beijing, Guangdong, Hunan, Fujian, and other provinces with a high level of coupled and coordinated development, but obviously they have not been driven by the neighboring Guangdong Province in the development process, and the spillover effect between regions has little influence on them. Therefore, in the future development process, exchanges and cooperation with the surrounding high-concentration areas should be promoted, the areas should improve their own strength by introducing the technology and experience of high-level areas, and the coordinated development of digitalization and economic growth should be promoted.

4.  High–low agglomeration area. In 2010, these areas included Liaoning, Hubei, Hunan, Guangdong, and Sichuan provinces; Beijing, Hubei, Guangdong and Sichuan also fell into this category in 2019. The characteristic of this category is that the coordination between digitalization and economic development in this kind of area is at a high level, but the coupling coordination level of neighboring provinces is weak. Due to the high level of self-development and the weak development level of neighboring provinces, the low-level areas have a certain polarization effect on the high-level areas, which inhibits the development of such areas to a certain extent, and the provinces with high coupling coordination do not fully promote the common development of neighboring provinces. Therefore, in future development, cross-regional cooperation and exchanges with neighboring provinces should be actively carried out to alleviate the imbalance of regional development. The radiation effect as the core of urban agglomeration should be fully exploited, and neighboring provinces should be driven to jointly improve the level of digitalization and economic development and realize comprehensive and coordinated development.

## 4. Conclusions

From the perspective of sustainable development, this paper discusses the relationship between digital transformation and economic development, establishes the evaluation system of digital transformation and economic development levels, and calculates a comprehensive index of digital transformation and economic development levels. Using the coupling coordination degree model, this paper explains the coupling coordination relationship between digital transformation and economic development in China, analyzes and discusses it in terms of time and space, and finally draws the following conclusions:

1.  From 2010 to 2019, the comprehensive index of digital transformation and the comprehensive index of economic development in 30 provinces and cities in China showed an overall upward trend. Except for Liaoning Province, the comprehensive index of digital transformation declined slightly in 2016, and the other provinces and cities showed a steady upward trend, but there was a large gap between the provinces and cities, which showed that the development level of the eastern, central, and western regions decreased.

2.  From the perspective of the coupling coordination degree, the coupling coordination degree of digital transformation and economic development in 30 provinces and cities in China increased year by year from 2010 to 2019, indicating that the interaction between the two systems has been continuously enhanced and the coordination degree between the two systems has been continuously improved. From the perspective of space, the degree of coupling coordination shows a decreasing trend from the eastern coast to the interior of the central and western regions. In terms of development types, except Guangdong Province, all provinces and cities in China exhibit digital lag coordinated development, which indicates that there is still much room for development in terms of the coordinated relationship between digital transformation and economic development.

3.  The degree of coupling coordination between digital transformation and economic growth in China is positively correlated on the whole, and there is a significant agglomeration effect in space. High–high agglomeration areas are mainly concentrated in Beijing–Tianjin–Hebei and Yangtze River Delta regions, while low–low agglomeration areas are concentrated in northeast and west regions, while low–high concentration areas and high–low concentration areas are concentrated in southeast provinces.

## 5. Contributions and Suggestions

At present, China is entering the critical moment of industrial structure adjustment, optimization, and upgrading. As a new direction of economic development, digital transformation plays an important role in China's economic construction. Therefore, exploring the internal relationship between digital transformation and economic growth is of great significance for promoting the synchronization of digital transformation and sustainable economic development. The theoretical and practical contributions of this paper are as follows:

1.  The synergy between digital transformation and economic development enriches the research into economic effectiveness driven by digital technology. The rapid growth of China's economic data and academic literature have proved that the digital transformation has effectively promoted the high-quality transformation and upgrade of China's economy. However, most of the existing studies have analyzed the mechanism of digital transformation to promote economic development through a large number of theoretical analyses and empirical models, and these one-way research works ignore the effective interaction between digital transformation and economic development. In this paper, based on the perspective of the synergy between digital transformation and economic development, the coupling and coordination relationship between digital transformation and regional economic development is discussed from the provincial level in China. The research results can provide a certain reference value for China's economy to move towards high-quality and sustainable development.

2.  Concerning our methodology, qualitative analysis and quantitative analysis are combined to obtain more scientific results and support more relevant studies for the two complex subsystems of digital transformation and economic development. Qualitative research methods are mainly used to find relevant literature, summarize the interaction mechanism, and select the system evaluation index. The quantitative research method employed was to collect the statistical data of 30 provinces and cities, use the entropy method to determine the index weight, and measure the level of digital transformation and economic development. We used the coupling coordination model and spatial autocorrelation model to analyze the coupling coordination degree and spatial agglomeration relationship between these factors and analyzed the changing trends and comparative differences between digital transformation and economic development from the two dimensions of time and space.

3.  In conclusion, this research provides a theoretical reference to correctly understand the development differences of different provinces and determine a further sustainable development direction. Firstly, the average coupling coordination degree between digital transformation and economic development is constantly rising, proving the existing two-way positive influence. Secondly, the development levels of the two indexes are not completely synchronized. The fact that digital transformation is lagging behind economic development, except in Guangdong province, shows that digital transformation in most provinces need to be enhanced. The gap in the coupling and coordination degree between different provinces indicates that special attention should be paid to the interaction and coordinated development among provinces in the future. Thirdly, considering national sustainability, the high coupling coordinated provinces should encourage the neighboring provinces through radiation as the low coupling coordinated provinces focus on their own digital infrastructure construction and digital investment and strengthen cooperation and exchanges with high coupling coordinated provinces to narrow the regional development gap.

This study shows that China's digital transformation is a step-by-step process, and China still has not completely solved the problem of the economic development balance between the east and the west; there is therefore still much room for efforts in the direction of sustainable development.

1.  China should focus on the digital transformation and economic development process of the central and western regions and realize the regional balanced development of the digital economy. Only by improving the digital level of regions can the country quickly realize the sharing of information resources among regions, narrow the digital divide, and improve the matching efficiency of resources, which is of great significance for coordinating the unbalanced development of regional economy and realizing common economic development.

2.  Attention should be paid to the influence of the digital transformation process on society. Many business models have been derived from the digital economy, and more jobs have been created, which has reduced the unemployment rate to a certain extent and effectively promoted the sustainable development of the region.

3.  Investment and research and the development of information technology should be increased to lay a solid foundation for digital transformation and sustainable development. At present, in the global digital wave, governments of all countries regard the digital economy as the key to promote sustainable economic development and improve international competitiveness and raise it to a national strategic level. In this situation, China has realized that mastering more advanced digital technology can gain it an advantage in international competition. Therefore, it is of great significance to constantly improve the digital infrastructure network and accelerate the digital transformation to improve China's economic level, enhancing international competitiveness and gaining opportunities for future development.

## 6. Limitations

This paper has some limitations. From the provincial perspective, this paper mainly analyzes the development level of digital transformation and the economy, as well as the coupling and coordination relationship between them. Finally, the spatial correlation and agglomeration characteristics among provinces are observed by the Moran index. In the future research, more detailed data from prefecture-level cities can be collected to carry out the coupling and coordination degree research. In addition, the influence mechanism and specific implementation path of the coupling coordination between digital transformation and economic growth are also further research directions.

**Author Contributions:** Conceptualization, M.Z. and R.L.; Data curation, M.Z., R.L. and D.D.; Formal analysis, M.Z., R.L. and D.D.; Investigation, R.L. and D.D.; methodology, M.Z., R.L. and D.D.; Project administration, M.Z.; Software, R.L.; Supervision, M.Z. and D.D.; Visualization, R.L.; Writing-original draft preparation, R.L.; and writing-review and editing, M.Z. and D.D. All authors have read and agreed to the published version of the manuscript.

**Funding:** This research was funded by the General Program of the MOE Layout Foundation of Humanities and Social Sciences (No. 17YJA880014).

**Institutional Review Board Statement:** Not applicable.

**Informed Consent Statement:** Not applicable.

**Data Availability Statement:** No new data were created or analyzed in this study.

**Acknowledgments:** We thank the academic editors and anonymous reviewers for their kind suggestions and valuable comments.

**Conflicts of Interest:** The authors declare no conflict of interest.

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
