# Peer review of "Synergistic Effect between China’s Digital Transformation and Economic Development: A Study Based on Sustainable Development"

_sustainability, doi:10.3390/su132413773_

Round 1

Reviewer 1 Report

Generally, this paper is well written. The topic is interesting. The materials are well presented. There are several things I would like to point out:

  1. On page 3, the paper talks about entropy method and lists 6 equations, without explaining what the parameters are. It is very hard for readers to follow.
  2. On page 5 in table 2, the sum of the weight from last column does not equal to 100%.
  3. On page 5, “data sources”, the paper should talk about the details of how to collect the data.
  4. The paper should provide a summary of descriptive table for the raw data collected.
  5. On page 10 first line, the paper says: “the Moran’s 1 index values were all greater than 0.25”. But all the numbers are smaller than 0.25.
  6. There are several type errors, such as on page 8, the third paragraph, the third line, 2017 should be 2019.
  7. At the conclusion part, the paper should emphasize more the contribution of this paper to the literature.

Author Response

请参阅附件。

Reviewer 2 Report

Dear Author:

The presented paper " Research on Coupling Coordination between Digital Transformation and Economic Development in China " analyze an interesting topic such as the level of transformation of 30 provinces in China and how this development affects and feeds back into economic growth. The analysis is considered adequate, however, some recommendations for improvement are made.

First of all, it is recommended that the links to the official websites mentioned be included: China Statistical Yearbook, China Torch Statistical Yearbook, China Science and Technology Statis-199 tical Yearbook, China Software and Information Service Industry Development Report, 200 China Information Society Development Report and National Bureau of Statistics.

On the other hand, an analysis of correlations between the different variables that measure digital transformation and economic development could be interesting. From this correlation, it would be possible to recommend a specific factor to be strengthened. Or justify why its inclusion is not relevant.

Finally, it would be interesting to develop in greater depth recommendations for the digital development of the most disadvantaged provinces, pointing out which variables of digital transformation could have a greater effect on economic growth.

Kind regards

Author Response

请参阅附件。
